# The Active Mechanism of Nucleosome Depletion by Poly(dA:dT) Tracts In Vivo

**DOI:** 10.3390/ijms22158233

**Published:** 2021-07-30

**Authors:** Toby Barnes, Philipp Korber

**Affiliations:** Biomedical Center (BMC), Divison of Molecular Biology, Faculty of Medicine, LMU Munich, 82152 Munich, Germany; Toby.Barnes@bmc.med.lmu.de

**Keywords:** nucleosome, chromatin, poly(dA:dT), remodeling, RSC, yeast, *S. cerevisiae*

## Abstract

Poly(dA:dT) tracts cause nucleosome depletion in many species, e.g., at promoters and replication origins. Their intrinsic biophysical sequence properties make them stiff and unfavorable for nucleosome assembly, as probed by in vitro nucleosome reconstitution. The mere correlation between nucleosome depletion over poly(dA:dT) tracts in in vitro reconstituted and in in vivo chromatin inspired an intrinsic nucleosome exclusion mechanism in vivo that is based only on DNA and histone properties. However, we compile here published and new evidence that this correlation does not reflect mechanistic causation. (1) Nucleosome depletion over poly(dA:dT) in vivo is not universal, e.g., very weak in *S. pombe*. (2) The energy penalty for incorporating poly(dA:dT) tracts into nucleosomes is modest (<10%) relative to ATP hydrolysis energy abundantly invested by chromatin remodelers. (3) Nucleosome depletion over poly(dA:dT) is much stronger in vivo than in vitro if monitored without MNase and (4) actively maintained in vivo. (5) *S. cerevisiae* promoters evolved a strand-biased poly(dA) versus poly(dT) distribution. (6) Nucleosome depletion over poly(dA) is directional in vivo. (7) The ATP dependent chromatin remodeler RSC preferentially and directionally displaces nucleosomes towards 5′ of poly(dA). Especially distribution strand bias and displacement directionality would not be expected for an intrinsic mechanism. Together, this argues for an in vivo mechanism where active and species-specific read out of intrinsic sequence properties, e.g., by remodelers, shapes nucleosome organization.

## 1. Nucleosomes Are Depleted In Vivo over Poly(dA:dT) Tracts in Many Species

Nucleosome occupancy is prominently low over homopolymeric poly(dA:dT) sequence tracts in a wide variety of species including *S. cerevisiae* and other yeasts, fly, worm, mouse and human cells [1,2,3]. Nucleosome depletion scales directly with tract length. Clear effects usually start with five and more dA:dT base pairs (bp) in a row and are increasingly diminished with the number of interruptions in the homopolymeric sequence. As most nucleosome mapping is based on limited digests with Micrococcal Nuclease (MNase) [4,5] and as MNase has a sequence preference for dA:dT-rich sequences [6,7], there is the concern that the depletion of nucleosome signal over poly(dA:dT) in MNase-seq and related techniques reflects a technical bias. Indeed, it is recognized since long [8,9,10,11] and sometimes explicitly controlled for (e.g., [10,11,12]) that nucleosomes with dA:dT-rich DNA are depleted faster during MNase digestion kinetics. Nonetheless, the depletion over poly(dA:dT) tracts in vivo is not an MNase artefact as it is mostly (see below) stronger than can be explained by this MNase bias and, more to the point, confirmed, at least for some species such as *S. cereivisae*, with MNase-independent methods like anti-histone ChIP with sonication [11,13], ChIP-exo [14], chemical mapping [15,16] and DNA methylation footprinting [17,18,19].

## 2. Nucleosome Depletion Is Functionally Important and Caused by Poly(dA:dT) Tracts

Poly(dA:dT) tracts and their associated low nucleosome occupancy are linked to functional genomic regions like promoters (Figure 1 and Figure 2, poly(dA)/poly(dT) and in vivo panels) [20,21,22,23,24] and replication origins [25]. As the assembly of protein complexes like the transcription or replication machinery on these DNA regions would be hindered by nucleosomes, it is functionally important that there are mechanisms that remove nucleosomes from these regions. One of these mechanisms involves poly(dA:dT) tracts as their direct causality was demonstrated in vivo by introduction or removal of such tracts and the corresponding decrease or increase, respectively, in DNA accessibility/nucleosome occupancy [21,22,26,27]. It was suggested that transcription terminators, which also contain poly(dA:dT) tracts, are nucleosome depleted in yeast. This was demonstrated not to be the rule but rather linked to close proximity of terminators and promoters at tandem genes [28,29].

## 3. Special Intrinsic Properties of Poly(dA:dT) Tracts Were Suggested to Cause Nucleosome Depletion In Vivo by a Nucleosome-Intrinsic Mechanism

Many biophysical studies established intrinsic properties of poly(dA:dT) tracts that deviate from those of generic DNA sequences [3]. These properties include a shorter helical repeat, a more narrow minor groove, more extensive hydration, more extensive base overlap within one strand, and lead overall to increased stiffness (higher deformation energy [36]) and correlate with higher energetic costs for the wrapping and twisting of DNA around the histone octamer during nucleosome assembly. These costs can be observed and quantified in competitive nucleosome reconstitutions where several DNA sequences compete for nucleosome assembly [37,38,39]. The classical reconstitution method is salt gradient dialysis (SGD). Purified DNA and histone octamers are initially mixed at high ionic strength, usually 2 M NaCl, where DNA and histones hardly interact. The salt concentration is decreased gradually or in a stepwise manner by dialysis or dilution, respectively, so that the electrostatic interactions between DNA and histones [40,41] begin to drive nucleosome assembly. Due to the slow salt dilution, there is a sufficiently long period during which histones and DNA experience an ionic strength around ca. 1 M NaCl. This salt regime allows repeated nucleosome assembly and disassembly, i.e., the histone octamers, more precisely the (H3–H4)_2_ tetramers, which assemble first on the DNA, can effectively equilibrate to their energetically preferred positions and avoid unfavourable sequences [42,43,44]. This procedure was extensively used to compare different DNA sequences for their relative nucleosome formation propensities and to select especially strong nucleosome positioning sequences, like the “Widom 601” sequence [37,38,44,45]. The outcome of such measures is usually referred to as the “intrinsic” nucleosome formation propensity or affinity of a DNA sequence with “intrinsic” referring to combined properties of the nucleosome constituents, DNA and histones. Poly(dA:dT) tracts score low in this regard [37] and lead to low nucleosome occupancy, for example, in SGD with whole genomes [31,32,46].

It should be noted that SGD-derived intrinsic properties are measured under non-physiological conditions as the histones equilibrate to their positions at around 1 M salt and become kinetically stuck at lower salt concentrations, i.e., cannot re-equilibrate to other positions, even if these were the thermodynamically preferred ones at lower salt [38,41,43,44]. Incubation of SGD-reconstituted nucleosomes at low salt but elevated temperature, like 40–60 °C, allows histone octamers to move again along the DNA (“thermal nucleosome sliding”) and to re-equilibrate to positions that are preferred under these conditions and that may be different from the positions equilibrated at high salt but lower temperature [47,48]. This illustrates how intrinsic nucleosome positioning preferences depend on the thermodynamic conditions. Unfortunately, there is no technique available that would allow histone octamers to equilibrate to their intrinsically preferred positions at physiological ionic strength and temperature without inclusion of other non-histone factors, which may again tweak thermodynamic preferences. Therefore, we do not know the purely intrinsic sequence preferences for nucleosome assembly under physiological conditions. Intrinsic preferences measured by SGD or similar techniques, like equilibrium measurements at physiological buffer and temperature conditions but in the presence of the histone chaperone Nap1 [49], have to be taken as operational approximations of the in vivo situation [50].

Nonetheless, there is a striking correlation between nucleosome depletion over poly(dA:dT) tracts in vivo and in genome-wide SGD reconstituted chromatin in vitro (Figure 1, Figure 2, Figure 3 and Figure 4, compare MNase-seq nucleosome occupancy values over poly(dA)/poly(dT) tracts in in vivo versus in SGD chromatin) [31,32,46]. This and alone this *correlation* has been (mis)taken as evidence that this nucleosome depletion in vivo is mechanistically due to poly(dA:dT) tracts intrinsically disfavouring nucleosome assembly [3,51,52]. This mechanism is called “intrinsic” in the sense of “intrinsic to the nucleosome” to signify that only the intrinsic properties of DNA and histone octamer interacting in the nucleosome, without thermodynamic influence of other factors, determine such nucleosome depletion. While intrinsic properties of DNA and histones may be measured by various techniques, we note that there is a semantic twist in the context of SGD reconstitutions as here intrinsic nucleosome properties and intrinsic nucleosome positioning/depletion mechanism are irrevocably linked. In SGD, the outcome of nucleosome positioning or depletion is defined to result from an intrinsic mechanism and at the same time thereby defines intrinsic properties. In other cases, e.g., in vivo, the outcome of nucleosome organization also always reflects the causative mechanism, but need not reflect the intrinsic properties of just the nucleosome as the mechanism may involve other factors, too. Nonetheless, a purely intrinsic mechanism is appealing also for the in vivo situation as it would involve the minimal number of factors and assumptions and thereby conform with Occam’s Razor principle. Nonetheless, again, also by principle, correlation need not reflect causality, i.e., the similar outcome of nucleosome depletion over poly(dA:dT) tracts in vivo and in SGD chromatin need not reflect the same causative mechanism. In the following we delineate the evidence against the intrinsic nucleosome depletion mechanism in vivo. We do not dismiss that special intrinsic properties of poly(dA:dT) tracts also exist and matter in vivo, but we argue that they by themselves do not constitute the mechanism for nucleosome depletion over poly(dA:dT) tracts in vivo.

## 4. In Vivo Nucleosome Depletion over Poly(dA:dT) Tracts Is Not Universal

As canonical histones are among the most highly conserved proteins, it may be expected that the intrinsic mechanism based on the intrinsic nucleosome disfavouring properties of poly(dA:dT) tracts should play out across species. Indeed, nucleosome depletion over poly(dA:dT) was observed in SGD using histones from several species, e.g., fly or chicken histones [31,46], and in many species in vivo [1,2,3]. However, there is the notable exception of *S. pombe*. Here, nucleosome depletion over poly(dA:dT) is far less pronounced than in *S. cerevisiae* and dA:dT-rich sequences are not enriched in linkers or nucleosome-depleted promoter regions but rather within nucleosomes close to the dyad [53,54]. Nucleosome depletion over poly(dA:dT) as based on MNase-seq measurements for *S. pombe* was not confirmed by chemical mapping [54]. For example, dA:dT-rich replication origins in *S. pombe* appeared nucleosome depleted in MNase-based but not in chemical mapping. This is a case of misleading MNase sequence bias. We wonder if we had ever heard of the intrinsic “genomic code for nucleosome positioning” [55] had *S. pombe* rather than *S. cerevisiae* been the main model species.

As another exception from a putatively universal intrinsic mechanism, introduction of a (dA:dT)_19_ tract into a well-positioned nucleosome at the *S. cerevisiae PHO84* promoter did not lead to removal of this nucleosome like otherwise seen upon promoter activation, but a clear DNaseI footprint was maintained and nuclease accessibility increased only moderately [56].

## 5. The Energetic Penalty for Incorporation of Poly(dA:dT) into Nucleosomes Is Not Very High

The manipulated *PHO84* promoter nucleosome is a counterexample regarding the misconception that poly(dA:dT) tracts cannot be incorporated into nucleosomes at all. There are many other examples in vivo where nucleosomes are positioned over poly(dA:dT) tracts, for example by the ATP depedent *S. cerevisiae* remodeler ISW2 [57] or at (dA:dT)-rich terminator regions in *S. cerevisiae*, which leads to MNase-sensitive nucleosomes [11]. Additionally, in vitro reconstitution of nucleosomes onto poly(dA:dT)-containing sequences is well possible [58,59], including even a crystal structure of a nucleosome with a (dA:dT)_16_ tract [60].

Not only can poly(dA:dT) tracts be nucleosomal in vitro and in vivo, the respective energetic cost is actually not as high as initially assumed. The more extreme differences between intrinsic affinities of histone octamers to DNA sequences as measured by competitive SGD in the Widom group were ~5000 fold, which corresponds to an energetic difference (ΔΔG = RT ln5000) at 30 °C of ca. 5 kcal/mol. This is not even half the energy of one ATP hydrolysis reaction (ca. 12 kcal/mol under physiological conditions [61]). Nucleosome dynamics are mediated in vivo by ATP dependent chromatin remodeling enzymes (remodelers) [62,63] that (dis)assemble, reposition (slide) and reconfigure (histone exchange) histone octamers on DNA. For some remodelers a step size of 1–2 bp per ATP hydrolysis was estimated [64,65,66], which amounts to at least hundreds of ATP hydrolyses for sliding a nucleosome along DNA even over short distances. Therefore, we suggest that the energetic differences of intrinsic sequence preferences for nucleosome formation are comparatively low in a remodeler-dominated energy landscape and may be easily overcome by ATP input.

Further, reassessement of energy measurements arrived at quite smaller energetic differences than before. If the 5S rDNA positioning sequence was taken as reference point, then the Widom 601 strong positioning sequence showed a difference of 0.7 kcal/mol (~3 fold higher affinity) in the Nap1-assembly method [67], which is less than the originally reported 2.9 kcal/mol (~124 fold higher affinity) derived by competitive SGD. Comparison of mononucleosomes containing poly(dA:dT) tracts taken from *S. cerevisiae* promoter sequences showed that they are less stable by at most 0.86 kcal/mol (~4 fold lower affinity) relative to 5S rDNA-nucleosomes in competitive salt step reconstitutions [59], i.e., by less than 10% of ATP hydrolysis energy. Note that the 5SrDNA reference itself is already a strong nucleosome positioning sequence. Therefore, the energetic differences between poly(dA:dT)-containing sequences and the average genomic sequences will be even smaller. Nonetheless, this does not preclude that they matter in the end.

## 6. Nucleosome Depletion over Poly(dA) Tracts by the Intrinsic Mechanism In Vitro in SGD Is Much Weaker Than by the In Vivo Mechanism

The caveat that MNase digestion bias (see above) may exaggerate the appearance of nucleosome depletion, also due to the generation of shorter DNA fragments that may be lost during sequencing library preparation, was not controlled for in SGD chromatin by MNase-independent methods so far. As MNase-dependent nucleosome mapping has to employ limiting digests, resulting occupancy values (peak heights, trough depths) depend on the chosen digestion degree, which is difficult to control for or normalize for comparisons between independent experiments [12,17]. This obscures the comparison of nucleosome depletion between different conditions, like in vivo versus in vitro chromatin, by MNase-based methods. Further yet, SGD chromatin can be prepared at arbitrary nucleosome densities, which also may affect the degree of nucleosome depletion over poly(dA:dT) tracts.

Here, we addressed these so far unresolved issues by using published absolute occupancy data obtained by MNase-independent DNA methylation footprinting (occupancy measurement via DNA methylation and high-throughput sequencing: ODM-seq, [17]). This technique monitors occupancies via the ratio of DNA regions accessible versus protected from methylation by CpG and/or GpC methyl transferases. The methyl transferase sequence specificity precludes direct monitoring of poly(dA:dT) tracts. Nonetheless, there are sufficiently many CpG and GpC sites even in poly(dA:dT)-rich promoter regions to detect nucleosome depletion (Figure 1, compare the two rightmost panels). As the ratio of accessibility versus protection is determined under saturating conditions, monitors an aliquot of the total (in contrast to MNase-based or other methods that score only accessible *or* inaccessible regions) and corresponds to an absolute measure, it allows direct comparison of occupancies across conditions. We compared on the one hand ODM-seq data for *S. cerevisiae* in vivo chromatin, averaged over five replicates, as well as for SGD chromatin reconstituted along the *S. cerevisiae* genome at two different nucleosome densities [17] with on the other hand published MNase-seq data of in vivo chromatin and for SGD chromatin at three different densities each (Figure 1 and Figure 2) [33]. As ODM-seq reports nucleosome occupancy over the whole nucleosome length, we plotted MNase-seq data as 147 bp extended dyads, too. For richer biological context, we subdivided the genes into the four groups of RP (ribosomal protein), STM (SAGA/TUP/Mediator regulated), TFO (transcription factor organized, especially by general regulatory factors like Reb1 and Abf1) and UNB (unbound by anything but the preinitiation complex) genes, following a recent categorization via comprehensive mapping of factor binding by ChIP-exo [34].

Classical in vivo +1 nucleosome-aligned heatmaps (Figure 1) and composite plots (Figure 2) show the known enrichment of poly(dA)/poly(dT) tracts in promoter versus genic regions as well as the stereotypical in vivo pattern of regularly spaced nucleosomal arrays downstream and the prominent nucleosome-free region (NFR) over the promoter just upstream of the +1 nucleosome in both the ODM-seq and the MNase-seq data for many genes. Promoter NFRs in vivo often coincide with poly(dA)/poly(dT) tracts, but there are also mechanisms that generate NFRs in vivo with few or even without poly(dA:dT) tracts like binding competition with transcription factors or general regulatory factors, especially at RP genes [2,68,69,70,71,72]. As noted before [34], enrichment or focus of poly(dA)/poly(dT) tracts at promoters is most pronounced for UNB and TFO genes and seen for only some STM and few RP genes (see also below, Figure 5). For the poly(dA)/poly(dT)-enriched promoters, the intrinsic mechanism of SGD chromatin assembly generates promoter NFRs in vitro, too, as seen in the MNase-seq data. As mentioned above, this correlation led to the idea that an intrinsic mechanism were also at work in vivo. However, this correlation is much weaker in the ODM-seq data that reports nucleosome occupancy in absolute terms and without the MNase sequence bias.

Nucleosome density in SGD chromatin had hardly an effect on the degree of nucleosome depletion in all cases. The corresponding traces of the three densities were virtually identical in the MNase-seq data (Figure 2, middle panel, Figure 3 and Figure 4, left and middle panels). As expected and shown before [17], absolute occupancy levels scaled with nucleosome density in the ODM-seq data (Figure 2, Figure 3 and Figure 4, right panels). The comparison between absolute occupancies of in vivo chromatin versus of SGD chromatin at nucleosome density of 0.8 showed that such high nucleosome density in our genome-wide reconstitution system almost reached the in vivo density.

To address the poly(dA)/poly(dT)-driven nucleosome depletion more directly, we binned all genomic poly(dA) tracts into three length categories (5 to 9, 10 to 14 and more than 15 dA in a row) and prepared composite plots of the same MNase-seq and ODM-seq data as in Figure 2, but aligned at the centers of these poly(dA) tracts (Figure 3). We also added another replicate for MNase-seq data. This clearly showed how nucleosome depletion over poly(dA) tracts, in vivo and for all nucleosome densities in vitro, increased with tract length. Note that composite plot traces became more rugged with decreasing number of instances as generally true. The differences between the replicates of MNase-seq data for SGD chromatin likely reflected the differences in MNase digestion degrees, which are notoriously diffcult to control exactly. Importantly, depletion in vitro was very similar as in vivo if monitored by MNase-seq, but much weaker in ODM-seq data. We conclude that the intrinsic nucleosome depletion over poly(dA) tracts is inflated by the MNase digestion sequence bias, which preferentially removes nucleosomes or other DNA regions that contain (dA:dT)-rich sequences. Nonetheless, long poly(dA) tracts showed clear nucleosome depletion in vitro also by ODM-seq, which confirms that the intrinsic depletion over poly(dA) tracts can be real. It is just much weaker than the in vivo depletion.

The discrepancy between nucleosome depletion in vivo versus in vitro was much more pronounced over promoter regions (Figure 2) than over genome-wide poly(dA) tracts (Figure 3) for both MNase-seq and ODM-seq measurements. This was not due to the 10–18% of promoters per gene group without poly(dA) tracts, where nucleosome depletion has to occur by a poly(dA)-independent mechanism in vivo, as this was still seen if we focused on poly(dA) tracts and compared promoter versus non-promoter regions (Figure 4). As this was still seen also for MNase-seq, which even inflates the intrinsic depletion over poly(dA) tracts, this suggests that there are mechanisms in vivo, especially at promoters, that enhance nucleosome depletion over poly(dA) tracts relative to the intrinsic mechanism in vitro (see below). In addition, this may reflect the influence of the known poly(dA)-independent mechanisms, like factor binding competition [68], at promoters. Again, in Figure 4, nucleosome depletion in vitro was in all cases much less pronounced than in vivo if monitored by ODM-seq.

Together, we confirm that there is poly(dA)-dependent nucleosome depletion by an intrinsic mechanism in vitro, and that there is a positive correlation with depletion over poly(dA) in vivo. However, the effects in vitro are so much weaker than in vivo if monitored by absolute occupancy in an MNase-independent way and especially in promoter regions enriched in poly(dA)/poly(dT) tracts (Figure 1, Figure 2 and Figure 4), that we wonder if anyone had ever drawn mechanistic conclusions from this correlation if MNase had not been the principal mapping tool. The enhanced depletion at promoters suggests additional, presumably active mechanisms as outlined in the following.

## 7. Nucleosome Depletion over poly(dA:dT) Tracts Is Not an Intrinsic Default State but Actively Maintained

If poly(dA:dT) tracts excluded nucleosomes in vivo by a merely intrinsic mechanism, one may expect that they do so without further maintenance. Indeed, most poly(dA:dT) tracts in *S. cerervisiae* promoters are constitutively nucleosome-free and there is no viable mutant known where this is not the case [34,73]. However, if the essential ATP dependent chromatin remodeling complex RSC and/or a certain class of essential sequence specific DNA binding proteins with binding sites in promoters, so called general regulatory factors (GRFs), like Reb1, Abf1 or Rap1, are ablated in conditional mutants, these NFRs start to fill up with nucleosomes [73,74,75,76,77,78,79,80,81,82]. This argues that promoter NFRs over poly(dA:dT) tracts are actively maintained by these factors and we suggest that this is the main reason why these factors are essential. Other non-essential remodelers, like the INO80 or the ISW2 remodeling complex, also participate in shaping promoter NFRs as NFR borders are affected in their respective deletion mutants [73,77,83,84] and in in vitro reconstitutions with these remodelers [32,33,35]. Further, NFRs with or without poly(dA:dT) tracts are assembled into nucleosomes in the wake of replication and only later become nucleosome-free again [85,86,87].

## 8. The *S. cerevisiae* RSC Remodeling Complex Preferentially Evicts Nucleosomes from Poly(dA:dT) Tracts

Active nucleosome depletion over poly(dA:dT) tracts could result from removing nucleosomes either in cis along the DNA (sliding) or in trans (disassembly). Turnover experiments in *S. cerevisiae* monitoring differentially labelled histones demonstrated nucleosome removal in trans, especially at promoter NFRs [88,89,90,91]. Nucleosome disassembly in trans is catalysed by only a subset of ATP dependent remodelers. Remodelers are classified according to sequence similarity among their ATPase motor subunits into several families, of which the four major ones are the SWI/SNF, INO80, CHD and ISWI remodelers [62,63,92]. Only members of the SWI/SNF family show nucleosome disassembly activity in vitro. *S. cerevisiae* contains two SWI/SNF type remodelers, the SWI/SNF complex, after which the family was named, and the RSC complex [92]. Both are involved in promoter NFR or NDR (nucleosome depleted regions, see comment on terminology below) formation in vivo, but only RSC has an essential and pervasive role [73,74,75,76,77,78,79,80,81,82,93]. Only RSC but not SWI/SNF was able to reconstitute in vivo-like promoter NFRs in the context of a RSC-depleted whole cell extract in vitro [94] and RSC showed a preference for remodeling promoter versus genic nucleosomes in ex vivo mini-chromosome circles [95]. This suggested that RSC may recognize some promoter feature. Indeed, RSC preferentially evicted histone octamers from poly(dA:dT) tract-containing DNA in mononucleosome assays in vitro [59].

## 9. Genomic Strand Bias of Poly(dA) Tracts and Directional Nucleosome Displacement from Poly(dA) Tracts Argues against the Intrinsic but for an Active Nucleosome Depletion Mechanism

The above arguments already point towards an active and targeted role for RSC in clearing nucleosomes from poly(dA:dT) tracts in vivo and in vitro. Nonetheless, they could still be regarded as circumstantial and at least qualitatively compatible with the suggested intrinsic mechanism of nucleosome depletion or with a mechanism where RSC or other remodelers may just implement the intrinsic sequence preferences. *S. pombe* in general [54] and *S. cerevisiae* promoters with ISW2-dependent nucleosome positioning [57] in particular may be examples where some other mechanism overrides mechanisms solely based on intrinsic biophysics. However, there are more arguments based on poly(dA) strand bias and nucleosome displacement directionality by RSC that clearly argue against the intrinsic mechanism.

Many *S. cerevisiae* promoters show a biased strand distribution of poly(dA)/poly(dT) tracts such that poly(dT) is often upstream and/or poly(dA) downstream of the NFR center on the coding strand (Figure 1 and Figure 2, left panels; Figure 5a,b) [32,96,97]. Such bias would not be expected to evolve in the context of a purely intrinsic mechanism. If in vivo nucleosome occupancy data are aligned at the center of poly(dA) tracts (Figure 3 and Figure 4, ODM-seq data) [96] or at the midpoint between pairwise combinations of 5′-poly(dA)-poly(dT), 5′-poly(dA)-poly(dA) or 5′-poly(dT)-poly(dA) tracts on the same strand [96], the NFRs are formed asymmetrically with a 5′ or 3′ offset relative to poly(dA) or poly(dT), respectively. De Boer and Hughes [96] noted that the 5′-poly(dT)-poly(dA) pairwise arrangement thereby leads to mutual reinforcement of nucleosome displacement (from both sides towards the center of the paired tracts, see schematic on the right of Figure 5c) and the strongest NFRs, which may explain why this evolved as a common promoter organization in *S. cerevisiae* [96,97]. Asymmetric NFR formation was also seen in mouse and human cells, although with opposite directionality, as well as in yeast genome-wide chromatin reconstitutions with whole cell extracts and ATP [96], arguing for a role of remodelers. Again, such asymmetry or strandedness of the poly(dA)-effect, especially in a species-specific way, would not be expected for a purely intrinsic mechanism. For intrinsic nucleosome exclusion it should not matter which strand has the A and which the T bases. Indeed, SGD chromatin does not show a 5′ offset of NFR formation over poly(dA) tracts shorter than 14 bp (Figure 3 and Figure 4) [96], which also controls that this 5′ offset in most cases is not due to a biased distribution of poly(dA:dT) tracts in the vicinity of the alignment point. Even though the 5′ offset in vivo was seen before in MNase-seq data [96], we note that it is mainly seen in ODM-seq but not in MNase-seq data (Figure 3 and Figure 4).

The directional nucleosome depletion over poly(dA) tracts in vivo suggested that promoter sequences evolved so that they fit to a nucleosome depletion mechanism that reads out poly(dA) tracts in a directional way, possibly via remodelers. Wu and Li [97] almost prophetically suggested (verbatim quote) “It is possible that G:C-capped poly(dA:dT) tracts may mark an initiation point for the remodeling activity of ATP dependent chromatin remodelers such as RSC.” Indeed, this mechanism (without taking dG:dC-capping into account) was directly demonstrated in our genome-wide chromatin reconstitution system using purified components only [32]. SGD reconstitution of nucleosomes onto a genomic plasmid library representing essentially the whole *S. cerevisiae* genome preformed promoter NFRs according to the intrinsic nucleosome exclusion properties of poly(dA:dT) tracts. Otherwise, such SGD chromatin is not very in vivo-like as it lacks, for example, regularly spaced nucleosomal arrays, unless remodelers with spacing activity are added [30,32,46,98]. Incubation of this SGD chromatin with purified RSC and ATP widened these NFRs substantially and in a directional way leading to a 5′-offset relative to poly(dA) and accordingly to a 3′ offset relative to poly(dT) in promoters. Only RSC showed this directional nucleosome displacement, not the SWI/SNF, ISW1a, INO80 or ISW2 remodelers, which also controls against an MNase-bias effect. Subsequent in vivo analyses that took poly(dA) tract directionality into account confirmed the corresponding directional effects on NFRs upon RSC ablation [78].

For the generation of promoter NFRs by such a RSC-mediated active and directional mechanism of nucleosome depletion over poly(dA:dT) tracts, it would be expected that promoters evolved poly(dT) tracts 5′ and/or poly(dA) tracts 3′ of the NFR centers, i.e., either in a one-sided or two-sided arrangement, but not the other way around (schematic on the right of Figure 5c for a two-sided arrangement, for all other arrangements see definitions underneath the x-axis in Figure 5c, left). Indeed, poly(dT) is more abundant 5′ than 3′ relative to the NFR centers and vice versa for poly(dA) (Figure 5a,b). More than half (55%) of UNB gene promoters, which contain at least one poly(dA) or poly(dT) tract, show an arrangement of poly(dA:dT) tracts that strictly conforms with an active and directional RSC-mediated mechanism, i.e., poly(dT) 5′ and/or poly(dA) 3′ but no poly(dT) 3′ and no poly(dA) 5′ of the NFR center. This percentage decreases in the order of UNB > TFO > STM > RP gene promoters (Figure 5c, rightmost group of bars) and thus confirms the conclusion by Rossi et al. [34] that especially UNB and TFO promoters evolved a mechanism for constitutive NFR formation by strand-biased poly(dA:dT) tracts.

Finally, RSC was literally “caught in the act” while disassembling nucleosomes from *S. cerevisiae* promoter regions in vivo. RSC complexes were extracted from yeast nuclei by the CUT&RUN technique and probed for their genomic location and also for histone content by anti-histone immunoprecipitation [99]. They were indeed enriched in promoter regions and associated with histones, especially the H2A histone variant H2A.Z, as well as with GRFs. This finding also contributed to the debate around “fragile nucleosomes” (FNs). FNs were operationally defined as particles of roughly nucleosome size that were detected in MNase-seq only at mild but not at more extensive MNase digestion degrees in yeast promoters [100]. Their nucleosomal nature was questioned as, for example, neither ChIP-exo nor chemical mapping detected histone-DNA contacts in these regions [11,14,15,16]. In the end, the nature of FNs is not fully resolved but in promoter regions they apparently correspond to non-nucleosomal factors, like transcription (co)factors, or to RSC complexes that are in the process of disassembling nucleosomes from (dA:dT)-rich regions and thereby prevent the detection of histone-DNA contacts. This underscores again that it is not the intrinsic default state of such regions to be nucleosome-free, but that there is a constitutive nucleosome assembly system at work, as also seen in the wake of replication, that is constantly and actively counteracted by the disassembly activity of RSC.

## 10. Remodelers Are Information Processing Hubs That Turn DNA Sequence Information into Nucleosome Organization

Together, this strongly argues that poly(dA:dT) tracts are specifically recognized by RSC so that RSC actively, preferentially [59] and directionally [32] displaces nucleosomes from these tracts [99]. This mechanism probably entails that RSC reads out some special properties of poly(dA:dT) tracts while translocating along its tracking strand in the 3′–5′ direction [101]. These properties may well be related to the undebated intrinsic properties of poly(dA:dT), but encompass also an important directional component and may emerge especially during RSC remodeling. Therefore, we do not argue against intrinsic *properties* of poly(dA:dT) tracts per se, but argue that nucleosome depletion over poly(dA:dT) tracts in vivo does not occur by an intrinsic *mechanism*, which entails only the interactions of DNA and histones as determinants of the resulting nucleosome organization. Such an intrinsic mechanism seems unlikely in the light of the arguments given above and would definitely not show a strand bias with regard to the distribution of poly(dA) tracts around NFR centers and not show a directional effect in the context of nucleosome displacement by RSC. Instead, the special poly(dA:dT) properties are actively implemented by extrinsic factors, for example, the RSC remodeler in *S. cerevisiae*. It remains to be shown, which protein domains in RSC or other remodelers or which other additional factors mediate the read out of poly(dA:dT). We note that we cannot exclude the formal possibility that an intrinsic mechanism is at work also in vivo in some yet to be identified cases.

In general, in vitro reconstitutions with purified remodelers demonstrated that remodelers like *S. cerevisiae* Chd1, RSC and INO80 are information processing hubs for nucleosome organization, with organization encompassing positioning as well as disassembly of nucleosomes. They are able to read out DNA sequence and other information, e.g., distance to barriers like GRFs, and process this information together with their own input into remodeler-specific nucleosome organization [32,35,102,103]. The remodeler-specific information was conceptualized as a “remodeler ruler” [33,104]. Especially for the INO80 complex, the modules that make up the ruler and are involved in sensing and relaying DNA sequence information, in particular DNA shape features [105], were recently defined in structural terms [33,35]. Additionally, recently, the concept of intrinsic DNA bendability was revisited by a novel genome-scale cyclizibility assay for nucleosome-sized DNA fragments [106]. Here, DNA sequence-intrinsic properties (bendability) correlated with in vivo nucleosome organization in *S. cerevisiae*, i.e., low bendability in NFRs and linkers versus high bendability in nucleosomes, so that a “DNA mechanical code” was suggested. Importantly, also these authors did not consider this an updated version of the purely intrinsic nucleosome positioning mechanism, but show, also with INO80 as example, that these differential DNA mechanics features are read out not just by the histone octamer but also by a remodeler.

All this fits to the earlier observation that different remodelers generate different nucleosome positioning on the same DNA sequence (“remodeler code” [107]) as they read out sequence information [108]. The DNA sequence information corresponds less so to classical sequence motifs but rather to the more redundantly implemented DNA shape and mechanics features. It should be noted that the classical intrinsic sequence preferences for nucleosome positioning as compared in competitive SGD reconstitutions represent rather static properties restricted to the contribution of DNA and histones. In contrast, remodelers may read out more dynamic DNA properties during twisting and bulging DNA in the context of the histone octamer and in the course of their ATP dependent remodeling mechanism [66,109].

## 11. Species-Specific Strategies for Nucleosome Depletion

Nucleosome organization by remodeler-specific input offers a versatile and regulatable mechanism that explains in vivo observations beyond the best studied *S. cerevisiae* model. The at first confounding sequence features of nucleosome organization in *S. pombe* may simply mean that *S. pombe* remodelers evolved different ways of reading sequence features. For example, the *S. pombe* RSC complex lacks the Rsc3/30 subunits [110] that confer DNA binding interfaces for *S. cerevisiae* RSC [75]. Additionally, other yeasts use poly(dA:dT) to varying degrees in their promoter organizations [2]. Further, poly(dA:dT) tracts are enriched in human promoters [111] but may not play the main role for nucleosome depletion there. Rather, human promoters that contain CpG islands, i.e., are rich in dG:dC bp, correspond to the constitutively open poly(dA:dT)-rich *S. cerevisiae* promoters [112]. The enrichment of dG:dC bp, i.e., “nucleosome favouring” sequences in the intrinsic SGD-derived sense, is typical for promoters of human and other multicellular species [113,114]. In contrast to *S. cerevisiae* where the vast majority of the genome is constitutively expressed and most promoters, especially of the UNB and TFO genes [34], have a hard-wired “open door policy” [115] via poly(dA:dT)-enriched promoter sequences, only a minor genome fraction is expressed in each individual cell of a multicellular organism. Therefore, it appears that mechanisms for constitutive nucleosome depletion are less often hard-wired by directly remodeler-recognized sequences, but rather that regulatable mechanisms of indirect remodeler-recruitment, e.g., via transcription factor binding sites or by recruitment via histone-modifcations, open promoter chromatin on demand. Recently, an in-depth in vivo analysis of promoter nucleosome organization in two different human cell types, each with and without cell-type-specific gene induction stimuli, demonstrated cell type-specific, induction-specific and even very transient promoter opening [116].

## 12. Afterthought on the Terminology of Nucleosome Depletion

For more than two decades, low nucleosome occupancy regions were mostly called nuclease hypersensitive sites or, more specifically, DNaseI hypersensitive sites [117,118,119]. This terminology is still in use and has the advantage of stating exactly the experimental observation, i.e., these regions are hypersensitive to nuclease digestion, and is therefore always appropriate as long as it fits the observation. Together with the first genome-scale mapping of nucleosome positions [20], Oliver Rando introduced the term nucleosome-free region (NFR). This term brings with it an interpretation of the observation, i.e., the low mapping signal is due to the absence of nucleosomes. Of note, this term encompasses two statements that are subject to debate: what is a nucleosome and does “free” really correspond to zero or just low occupancy? Accordingly, the designation as NFR has received criticism, for example, in the context of the discussion about FNs or non-canonical nucleosome particles in promoter regions [100,120]. Many colleagues feel uneasy about the NFR term as it conveys a too extreme case of zero occupancy and they therefore prefer the more gradual connotation that comes with the term nucleosome depleted region (NDR), which is, to our perception, more pervasively used nowadays. Then, again, there is the mechanistic NFR-versus-NDR-distinction by Frank Pugh, who underscores that at least in *S. cerevisiae* there are truly nucleosome-free promoter NFRs as measured by ChIP-exo and chemical mapping [14,34,121], which are hard-wired as constitutively nucleosome-free. In contrast, NDRs in Pugh’s terminology correspond to regulated regions, e.g., *S. cerevisiae* promoters, which are covered with nucleosomes in their repressed state and depleted of nucleosomes in the course of promoter activation. We realize that this mechanistic distinction of NFR versus NDR seems to be less heard than the quantitative distinction and is used less often. Nonetheless, we follow the Pugh terminology here and mostly write NFR as our figures and analyzes refer to wild type yeast grown under standard growth conditions and as our arguments refer to nucleosome depletion by poly(dA:dT) tracts, i.e., we mostly refer to constitutive NFRs in the Pugh sense. In general, we are aware that observation and interpretation are epistemiologically entangled, but advocate to first use observational terms and then to add the interpretation by qualifiers. (Nuclease) hypersensitive site is an operationally defined term that is probably easy to agree on. Such hypersensitive sites can then be qualified and quantified as constitutive, regulated, nucleosome-free or else.

## Figures and Tables

**Figure 1 ijms-22-08233-f001:**
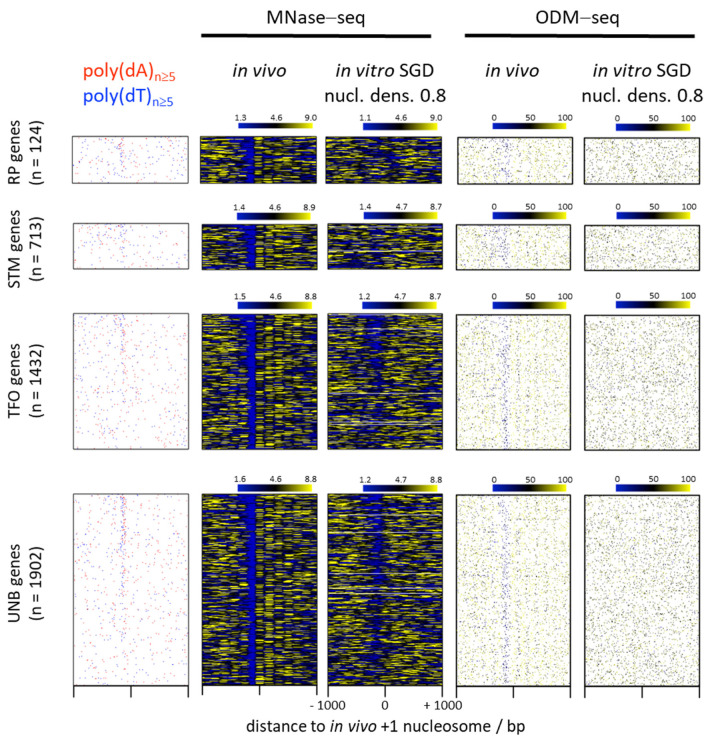
Correlation of poly(dA)/poly(dT) tract occurrence with low nucleosome occupancy in in vivo chromatin or in in vitro salt gradient dialysis (SGD) reconstituted chromatin monitored by MNase-seq or by DNA methylation footprinting (ODM-seq). Intrinsic nucleosome depletion over poly(dA)/poly(dT) tracts in SGD chromatin relative to the in vivo depletion in *S. cerevisiae* is less pronounced if monitored by ODM-seq than previously seen by MNase-seq [30,31,32]. From left to right: heatmaps (linked rows) of poly(dA)/poly(dT) tract occurrence on the coding strand, MNase-seq and ODM-seq data, each of in vivo (*S. cerevisiae* wild type BY4741) chromatin and of genome-wide in vitro salt gradient dialysis (SGD) chromatin reconstituted at a nucleosome density (nucl. dens.) of 0.8 (as defined via histone-to-DNA mass ratio, [33]). Data are subdivided into the groups of RP (ribosomal protein), STM (SAGA/TUP/Mediator regulated), TFO (transcription factor organized, especially by general regulatory factors like Reb1 and Abf1) and UNB (unbound by anything but the preinitiation complex) genes as defined in [34]. Number (n) of genes considered in each group is indicated. MNase-seq nucleosome dyad densities were normalised so that the sum of each row (gene) equals 1 [17,33] and plotted as 147 bp extended dyads. MNase-seq colour scales report normalized dyad densities ×1000 and range from the 10th to 90th percentile values of the individual panel, with extreme values outside these bounds being limited to the minimum/maximum of the scale. ODM-seq heatmaps report absolute nucleosome occupancy values ranging from 0% to 100%. Coordinates with missing values form a white background. All heatmaps are aligned at in vivo +1 nucleosome positions [35] and sorted in descending order from top to bottom by the number of bp within homopolymeric poly(dA) and poly(dT) tracts ≥5 bp long in promoter NFRs (nucleosome free regions between the borders (dyad position+ or −73 bp, respectively) of −1 and +1 nucleosomes [16]). Homopolymeric poly(dA)/poly(dT) tracts of at least 5 bp length were called by determining nucleotide frequency on the sense strand in a 5 bp sliding window with 1 bp step size. Every bp coordinate that is at the center of such a 5 bp homopolymeric poly(dA)/poly(dT) window is coloured in red or blue, respectively, all others in white. Ten percent, 14%, 18% and 14% of RP, STM, TFO and UNB gene promoters, respectively, have no poly(dA) or poly(dT) tracts ≥5 bp long within their NFRs. These genes are sorted in ascending order from top to bottom by genomic coordinate. MNase-seq data are from [33] of in vivo (GSM4175394) and in vitro (GSM4175430) chromatin. ODM-seq data are from [17] of in vivo (GSE141051) and in vitro (GSM4193216) chromatin.

**Figure 2 ijms-22-08233-f002:**
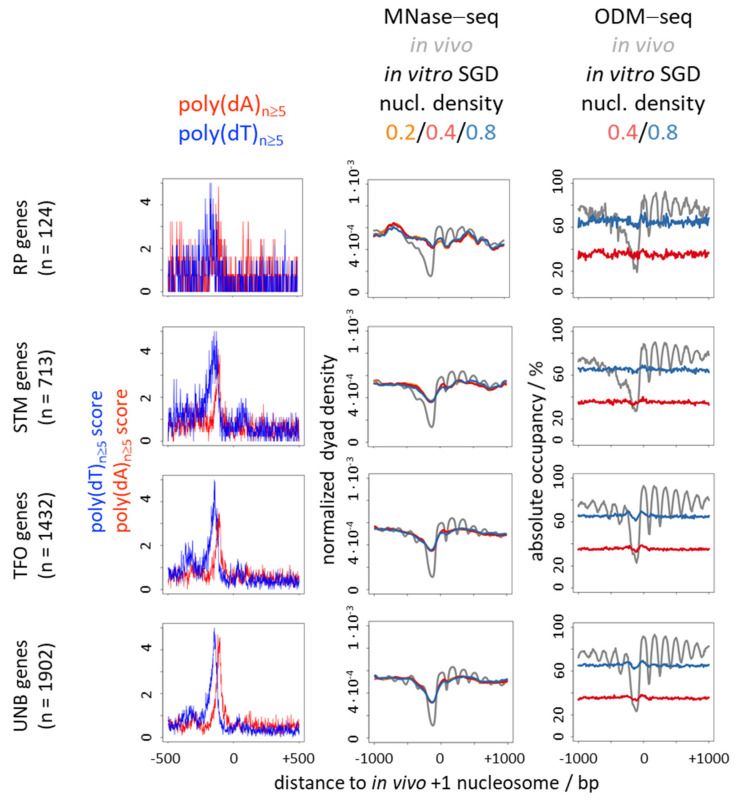
Alternative visualization of the strand bias of poly(dA) versus poly(dT) distribution in promoter regions as well as of the much less pronounced intrinsic nucleosome depletion over promoter regions in vitro compared to in vivo if monitored by ODM-seq versus MNase-seq. Relative nucleosome depletion does not depend on nucleosome density in in vitro reconstituted SGD chromatin. Composite plots of the same data as in Figure 1, and in addition of MNase-seq data from [33] of in vitro SGD chromatin with nucleosome density 0.2 and 0.4 (GSM4175428, GSM4175429, respectively) and of ODM-seq data from [17] of in vitro SGD chromatin with nucleosome density 0.4 (GSM4193222). Poly(dA)/poly(dT) scores correspond to the percentage of promoters in each group that have at the respective position along the x-axis a center of a homopolymeric poly(dA)/poly(dT) 5 bp window. Note that this gives an accurate distribution of poly(dA)/poly(dT) tracts, but underrepresents the number of bp within tracts as the outermost 2 bp flanking each tract are not represented.

**Figure 3 ijms-22-08233-f003:**
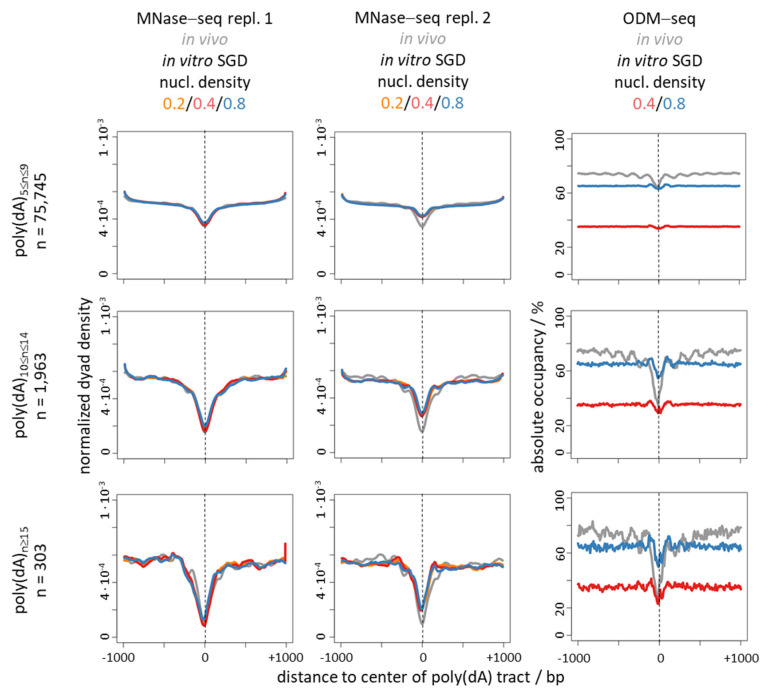
Nucleosome depletion over poly(dA) tracts scales with tract length and seems similar in extent in vitro versus in vivo if monitored by MNase-seq but not by ODM-seq. Composite plots of same data as in Figure 2 (MNase-seq repl. 1 and ODM-seq) plus an additional MNase-seq data replicate (MNase-seq repl. 2) for in vitro SGD chromatin with nucleosome density 0.2/0.4/0.8 from [33] (GSM4175803/GSM4175804/GSM4175805, respectively). Data are aligned at all poly(dA) tracts in the *S. cerevisiae* genome subdivided into the indicated tract length ranges. Number (n) of instances is indicated. Strand orientation was taken into account by flipping the orientation for poly(dA) tracts on the opposite strand. Vertical dashed lines mark the alignment points.

**Figure 4 ijms-22-08233-f004:**
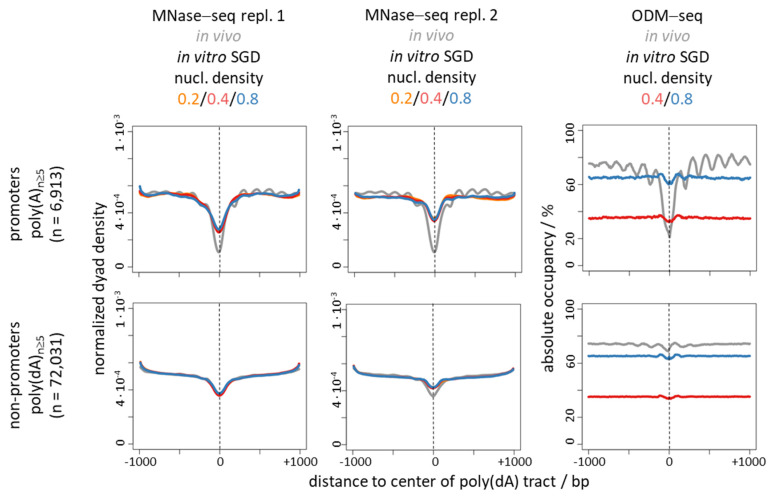
The difference between nucleosome depletion over poly(dA) tracts in vivo versus in vitro is much more pronounced in promoter versus non-promoter regions. Composite plots of the same data as in Figure 3, but aligned at all poly(dA) tracts ≥5 bp long within or outside of promoter NFRs [16]. Number (n) of instances is indicated. Strand orientation was taken into account by flipping the orientation for poly(dA) tracts on the opposite strand. Vertical dashed lines mark the alignment points.

**Figure 5 ijms-22-08233-f005:**
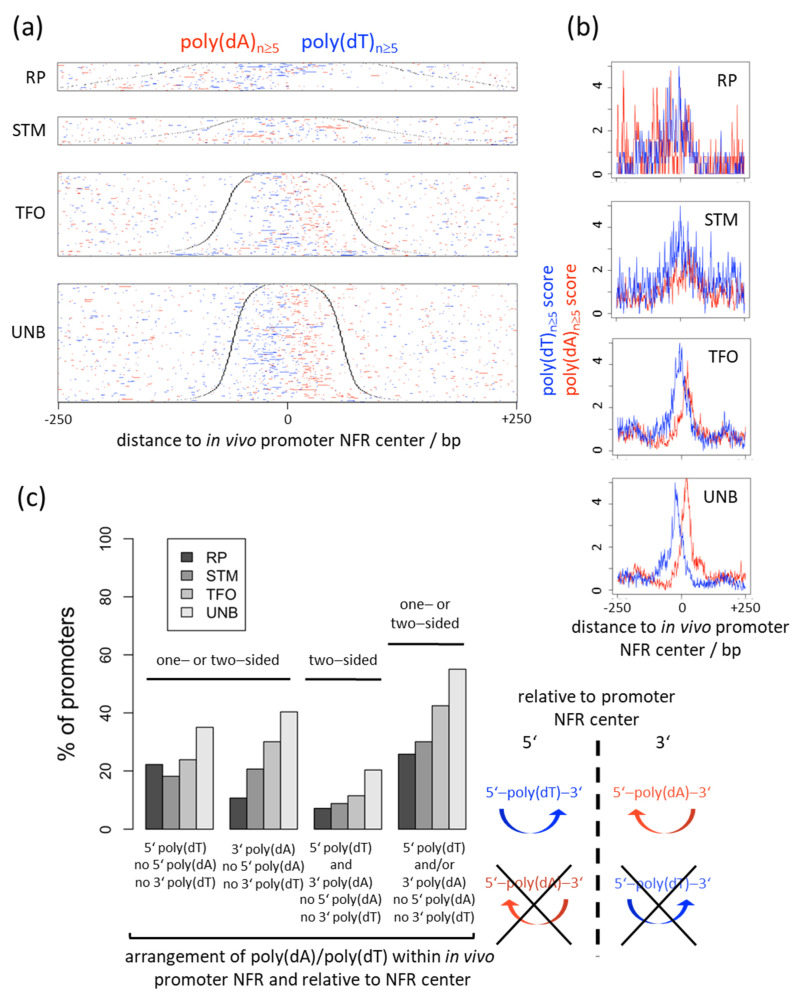
*S. cerevisiae* promoters evolved a strand-biased distribution of poly(dA) vs. poly(dT) tracts relative to their NFR centers that conforms with an active and directional RSC-mediated nucleosome depletion mechanism. (**a**) Heatmap as in Figure 1, leftmost panel, but aligned at NFR centers [16] and sorted from top to bottom by increasing NFR length. Black dots mark the downstream and upstream border of the −1 and +1 nucleosome, respectively. (**b**) Composite plots as in Figure 2, leftmost panel, but aligned at NFR centers. (**c**) Left: the percentage of promoters in the indicated gene groups that have at least one poly(dA) or poly(dT) tract of length ≥5 bp in their NFR and conform with the one- or two-sided arrangement of poly(dA)/poly(dT) in their NFR relative to the NFR center indicated below the x axis. Right: schematic of poly(dA)/poly(dT) tract arrangements around the NFR center (dashed vertical line) that conform or do not conform (crossed out) with the RSC-mediated active and directional nucleosome depletion mechanism over poly(dA)/poly(dT) tracts. Note that even a one-sided arrangement suffices to contribute to NFR generation by the active and directional mechanism. Curved colored arrows symbolize the directionality of nucleosome displacement by RSC relative to the equally colored poly(dA)/poly(dT) tracts.

## Data Availability

MNase-seq and ODM-seq data used in Figure 1, Figure 2, Figure 3 and Figure 4 are published in [17,33] and available at GEO under the accession numbers given in the respective figure legends.

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
