# Peer review of "The Active Mechanism of Nucleosome Depletion by Poly(dA:dT) Tracts In Vivo"

_ijms, 2021, doi:10.3390/ijms22158233_

Round 1

Reviewer 1 Report

ijms-1206477-peer-review

“The active rather than intrinsic mechanism of nucleosome 2 depletion by poly(dA:dT) tracts” by Toby Barnes and Philipp Korber.

The present manuscript argues for the hypothesis that observed nucleosome depletion over polyA tracts is the result of active mechanisms rather than purely a function of DNA-nucleosome interaction kinetics. The main argument is that the difference in computed binding energy is too small to account for the differences in observed and predicted occupancy and that therefore an active (ATP consuming) principle is involved in vivo. As outlined below, a less controversial formulation of the title and text, where both intrinsic and extrinsic types of mechanisms are proposed to be at play rather than (mistakenly, and in fact paradoxically as the authors acknowledge themselves) excluding intrinsic DNA properties altogether, as the title suggests.

Note: As author P. Korber is author on at least two manuscripts in press, it is possible that this manuscript does not include anything new relative to those papers (references 62 and 64). Alternatively, this is the forum where the authors can extensively make their arguments. The review is written with the latter scenario in mind.

Major Comments:

1. active versus intrinsic … nucleosome prediction tools on DNA bendability worjk very well.  This is a valid counterargument! But, as the authors do acknowledge, DNA intrinsically adopts conformation as a function of the nucleotide sequence. The Van der Heijden 2012 et al prediction tool (PMID: 22908247) is very sensitive to the addition of 19 Adenosines at position -560, in the middle of the very strongly positioned Pho84 promoter. The predicted occupancy likelihood drops almost 1 log (0.023 to 0.0037). In that model, the only reason for depletion is due to the length of this track which imposes multiple major groove AA and TT  base intra-strand Vanderwaals interactions across the di-nucleotide base steps in the parts of the major groove that face away from the surface of the nucleosome. It is an intrinsic property of the DNA sequence! This argues that intrinsic DNA deformability does dictate preferred nucleosome positions. While acknowledging this here and there in their arguments, the title of the paper is misleading in this respect. Essentially, the word ‘rather’ is misplaced in the title. A much less controversial title would be: “Active as well as intrinsic mechanisms involvement in nucleosome depletion by poly(dA:dT) tracts”

2. The alternative explanation that the authors do consider but also partially reject, is that SNF2-type nucleosome remodelers function in the context of the energetic landscape of DNA, and that the positioning of the remodelled nucleosomes rcan be predicted by the intrinsic properties of the DNA double helix that would be wrapped around the nucleosome. Data from references 102 and 103 in this manuscript support this view on the matter.

Remodelers: What protein protein domains bind to polyA tracts in promoters in yeast? The SNF2 family, and in particular RSC and Isw1/2 complexes are important for TSS-associated deviations from a thermodynamic model (see eg; van der Heijden et al 2012.) However, the known Rsc3/30 motif is CCCGCGCG (eg: Tsankov et al 2011), not a run of polyA and to my knowledge there is no other sequence-specific DNA-binding activity described for the yeast RSC complex subunits. Perhaps a discussion on the possible role of ARID domains that are known to bind to A:T-rich DNA may be useful for the readership?

Related to this; the author do not appear to have considered the potential importance of polyA:T binding proteins that are not associated with a SNF2-type remodeler?

3. Figure 1 and accompanying text: Are the authors confounding the issue by using “promoter versus non-promoter’ comparisons instead of ‘polyA tract UAS/promoters’ versus ‘non-polyA tract UAS/promoters’ ??? This can be alleviated by adding to figure 1 panels a-b-c also panels d-e-f for polyA>15 (ie; only instances with 15 or more consecutive A’s AND panels g-h-i only instances with 7 or less poly A tracts. For clarity the promoter versus non=-prmoter comparison is useful. However, the argument would be about promoters with and without polyA tracts and whther those without polyA tracts show less depletion of their promoters?

Minor:

a. Although poetic, the use of ‘ripples’ at the bottom of page 3 is not an appropriate term. A quantitative estimate would be expected. By contrast, the 10bp periodicity of preferred predicted and observed (vitro and vivo!) nucleosome positions in the genome is well-described as a ‘10bp rippling’ phenomenon, especially well-documented on the natural lytechinus variegatus 5S sequence. What SNF2 complexes that move nucleosomes do, is to conform or not to those ‘rippling’ instructions encoded by the DNA sequence in the form of stacking-interaction potential between adjacent bases of each DNA strand. Another type of ‘ripples’ concerns the nucleosomes that are positioned along chromosome segments as a consequence of actively positioned -1 and +1 nucleosomes at promoters.

b. The energy freed upon hydrolysis of the gamma phosphate of ATP provides an upper bound for the work that an ATPase may deploy per hydrolysis. However, it more than likely that this energy is distributed over many topologically elastic covalent interactions that initiate with ATP hydrolysis, all the way to release of the inorganic phosphate ion by the ATPase, in a cycle that takes more than 10 milliseconds to complete (Vmax for these ATPases is around 1000 ATP per minute). Indeed, pure Mi2 enzyme cannot move a nucleosome out of the Widom 601 sequence, suggesting that the required processivity and mechanical energy are not provided by ATP-hydrolysis by Mi2 on the 601 sequence (ref 103). Therefore, it is not intrinsic to a SNF2-ATPase to be able to use the amount of freed energy upon ATP hydrolysis to move any nucleosome over any sequence for a given distance! As the authors of references 102 and 103 concluded, in combination with their intrinsic protein- and DNA-binding properties, the remodelers do appear to follow DNA-intrinsic rules when catalyzing nucleosome movements along DNA, in vitro and therefore most likely also in vivo.

c. Provide real numbers! It is infuriating to read non-quantitative statements such as ‘Promoters are known to be especially enriched in polyA tracts…’. How many promoters? What are the criteria for ‘promoter’ and ‘polyA’ tract??? See major point 3.

d. Figure 3c has little informative value for this question as the two type of promoters, those enriched and those not enriched are neither named nor plotted! See remark on figure 1. Figure 3C does indicate that also ODM detects the depletion of nucleosomes over polyA tracts. That is a nice confirmation!

e. Page 10 “Remodelers are classi- 290 fied according to sequence similarity among their ATPase motor subunits into four major 291 families, the SWI/SNF, INO80, CHD and ISWI remodelers [56, 57, 89].” Vugt and Ranes BBA-GRM and other workers also include yfr038w/DDM1 as a distinct class of ATP-dependent SNF2-type nucleosome remodeling enzymes  (see PMID: 33833428)

f. Page 11 “Finally, 350 RSC was literally “caught in the act” while disassembling nucleosomes from S. cerevisiae 351 promoter regions.” Why talk about dissembling rather than sliding? The disassembly model has mainly been proponed by Y. Lorch and is less parsimonious than the sliding model since it requires acceptor molecules to which the nucleosome is transferred upon ‘dissasembly’. Most single molecule studies that I know of indicate sliding activity for RSC.

g. Data Availability Statement: accession numbers are only provided in the legend of figure 1, not 2 and 3.

h. The discussion on NFR versus NDR is good to review here, but it may be best placed initially, or at least before section 4. I would prefer it as a balanced statement in the introduction, rather than as a controversy that still needs to be resolved. As the authors note and write-up very well, the field has moved on.

i. The short discussion on fragile nucleosomes on page 11 on the other hand is not settled fully as it may well be a function of histone variant incorporation, as suggested by Barnes and Korber, but this is not yet causally demonstrated? If it is, it could be discussed in the introduction rather than in the results section?

Reviewer 2 Report

This paper is part bioinformatic analysis of published data, part review. It deals specifically with the question of the role of poly(dA) tracts in determining chromatin organisation. It has long been accepted by most that poly(dA) tracts play an important role in nucleosome depletion at budding yeast promoters. Personally, I have always thought that their role is obviously exaggerated in the literature. This paper provides a full discussion of this question and represents a valuable contribution to the literature. I agree with most of the authors' arguments; however, I do have a few comments:

  1. The figures presented here for chromatin reconstituted in vitro show that the intrinsic bias against nucleosome formation on poly(dA) tracts is really very small, except perhaps for relatively long tracts, which are quite rare. This is a major point made by the authors and I agree with it. However, given that this effect is so small, is there actually any compelling evidence for an important role of poly(dA) in chromatin organisation? Almost all of the studies cited here are correlative; direct proof is lacking - e.g., determining the effects of deletion or disruption of poly(dA) tracts on nucleosome positioning.
  2. In section 6 (page 4), ODM-seq data are introduced. It would be helpful if the authors give a very brief summary of the technique. They should also comment briefly on the fact that this method uses CpG/GpC DNA methylases and so cannot report directly on poly(dA) tract occupancy.
  3. Figure 2: MNase is biased toward AT-rich sequences, but it also produces short fragments that tend to be lost during library preparation. This effect might also contribute to deeper MNase-seq troughs than ODM-seq troughs (in which the DNA is not fragmented until after purification, if at all) at poly(dA) tracts.
  4. There is some evidence that poly(dA) tracts are involved in transcript termination by RNA polymerase II. This possibility deserves a mention.
  5. Line 355: "This finding also resolved the debate around “fragile nucleosomes” (FN). FNs were operationally defined as particles of roughly nucleosome size that were detected in MNase-seq only at mild but not at more extensive MNase digestion degrees in yeast promoters [96]." This statement is incorrect. The key finding in ref. 11 is that MNase-sensitive complexes observed at promoters do not contain histones. The more recent data (ref. 96) are inconsistent, because they are arguing for a RSC-histone complex.
  6. Paragraph beginning on line 397: It's worth pointing out that chromatin reconstituted in vitro is very different from chromatin in vivo, because it lacks regular nucleosome spacing. Nucleosome spacing enzymes are required.

Minor comments:

  1. Abstract: "5) S. cerevisiae promoters evolved a biased poly(dA) versus poly(dT) distribution." This statement should really have the word "directional" in it, since poly(dA) and poly(dT) are otherwise equivalent.
  2. Legend to Fig. 1: "Monitoring nucleosome depletion over poly(dA:dT) tracts by DNA methylation footprinting (ODM-seq) shows that intrinsic nucleosome depletion in SGD chromatin is even less pronounced relative to the in vivo depletion in S. cerevisiae than previously seen by MNase-seq." This figure does not examine the role of poly(dA) tracts at all.
  3. Line 223: "inflated by the MNase digestion sequence bias, which preferentially removes nucleosomes that contain (dA:dT)-rich sequences." This bias also applies to non-nucleosomal DNA, as in NDRs or linkers.
  4. Legend to Fig. 2: Typo "...in vivo versus in vivo....".

Reviewer 3 Report

This is a useful review, which will enhance the general understanding of the basis for nucleosome free region (NFR) formation. The authors make an important point about methodology, the bias introduced by micrococcal nuclease, with its strong propensity for attacking AT-rich DNA. One of their main conclusions, that NFRs are a product of RSC activity rather than the intrinsic properties of AT-rich DNA, has been clearly stated by Lorch et al. (2014) and further elucidated by Henikoff and collaborators. The authors should explicitly acknowledge those prior contributions in the first paragraph of p. 12. The point about the energetic cost of assembling a nucleosome on dA:dT DNA was also documented by Lorch et al. (2014). The important work of Madhani and coworkers on the role of a very short dA:dT sequence, which provides a powerful argument against intrinsic nucleosome exclusion, is ignored altogether.
The statement that the intrinsic properties of dA:dT DNA play a role in RSC-mediated NFR formation (first paragraph of p. 12) unfortunately muddies the picture the authors create and flies in the face of their well stated argument regarding correlation and causation (bottom of p. 2, top of p. 3). The lengthy discussion on p.12 of remodelers as "information processing hubs..not just enzymes" seems overlong and to state the obvious.
Typos:
line 155, "No" should read "Note"
line 459, "by" should read"be"

Round 2

Reviewer 1 Report

The authors suitably addressed the concerns of the reviewers. Still, some kind of tunnel vision appears to afflict them, as the new figure 1 clearly shows that the polyA stretches in question, be they blue or red cannot be the dominant source of nucleosome re-positioning or dissasembly, as a minority of promoters of any class harbour the stretches in question.

In fact, figure 1 does not display the 'regulatory' polyA strand bias that is central to the whole argumentation, or did I miss that, somehow? If so, please check the manuscript to make it more clear.

Perhaps the authors could be quantitative about the polyA stretches they analysed and state for how many promoters (of the four classes), a polyA [followed by a polyT?] stretch is putatively key to generate their NDR.

Author Response

We appreciate this second round of comments and suggestions as they prompted us to make the important aspect of strand-biased poly(dA)/poly(dT) tract distribution more clear in our manuscript. Alluding to the tunnel vision metaphor we hope that there is now light at the end of the tunnel.

Comments and Suggestions for Authors

The authors suitably addressed the concerns of the reviewers. Still, some kind of tunnel vision appears to afflict them, as the new figure 1 clearly shows that the polyA stretches in question, be they blue or red cannot be the dominant source of nucleosome re-positioning or dissasembly, as a minority of promoters of any class harbour the stretches in question.

We apologize that our Figure 1 apparently did not show this clearly enough. This is probably because of the rather zoomed out x-axis scale (-1 kb to +1 kb). At this scale, even long tracts appear as dots. While we would like to keep up this scale in Figures 1 and 2 to better visualize the genic nucleosomal arrays, we do add now a much zoomed-in version (-250 kb to +250 kb) in our new Figure 5a, where the tract lengths are now clearly visible as well as their position within the promoter NFRs. In addition, we added to the legend of Figure 1 the quantification that “Ten percent, 14%, 18% and 14% of RP, STM, TFO and UNB gene promoters have no poly(dA) or poly(dT) tracts ³ 5 bp within their NFR.”, i.e., >80% of promoters in all gene groups have such tracts.

In fact, figure 1 does not display the 'regulatory' polyA strand bias that is central to the whole argumentation, or did I miss that, somehow? If so, please check the manuscript to make it more clear.

Yes, while present, the strand-bias is a bit difficult to see in the leftmost heatmap of Figure 1. We thought that it would be more clear in the leftmost composite plot of Figure 2, at least for the UNB and TFO genes, which show this strand-bias most clearly.

Perhaps the authors could be quantitative about the polyA stretches they analysed and state for how many promoters (of the four classes), a polyA [followed by a polyT?] stretch is putatively key to generate their NDR.

Thank you for this excellent suggestion. We do this now in our new Figure 5c. In this context, we realized that we were not explicit enough about what kind of poly(dA)/poly(dT) arrangement is “putatively key” to generate an NFR via the active mechanism. We added now a whole section that states this explicitly (lines 449-462: “For the generation of promoter NFRs by such a RSC-mediated active and directional mechanism of nucleosome depletion over poly(dA:dT) tracts, it would be expected that promoters evolved poly(dT) tracts 5’ and/or poly(dA) tracts 3’ of the NFR centers, i.e., either in a one-sided or two-sided arrangement, but not the other way around (schematic on the right of Figure 5c for a two-sided arrangement, for all other arrangements see definitions underneath the x-axis in Figure 5c, left). Indeed, poly(dT) is more abundant 5’ than 3’ relative to the NFR centers and vice versa for poly(dA) (Figure 5a,b). More than half (55%) of UNB gene promoters, which contain at least one poly(dA) or poly(dT) tract, show an arrangement of poly(dA:dT) tracts that strictly conforms with an active RSC-mediated mechanism, i.e., poly(dT) 5’ and/or poly(dA) 3’ but no poly(dT) 3’ and no poly(dA) 5’ of the NFR center. This percentage decreases in the order of UNB>TFO>STM>RP gene promoters (Figure 5c, rightmost group of bars) and thus confirms the conclusion by Rossi et al. [34] that especially UNB and TFO promoters evolved a mechanism for constitutive NFR formation by strand-biased poly(dA:dT) tracts.”) and added a schematic that illustrates this (Figure 5c, right). We also realized that we generated the impression in our previous versions, that poly(dA) and poly(dT) should show a two-sided arrangement for NFR generation. However, this need not be the case, as already a one-sided arrangement supports the active and directional RSC mechanism as long as no poly(dA:dT) tract is present in the wrong direction. This is now stated explicitly in the section cited above (lines 451-452: “...i.e., either in a one-sided or two-sided arrangement...”) and in the legend to Figure 5c (“Note that even a one-sided arrangement as defined for the two bar groups in the graph on the left suffices to contribute to NFR generation by the active and directional mechanism.”) In addition, we added now heatmaps (Figure 5a) and composite plots (Figure 5b) aligned at the NFR centers as it is the arrangement of poly(dA)/poly(dT) tracts relative to the NFR center that matters most for the role of these tracts in generating the NFRs. For calling promoter NFR lengths and centers we used the chemical mapping data by Chereji et al (ref. 16) and, for consistency, also used this data for heatmap sorting in Figure 1 and for calling promoter poly(dA) tracts in Figure 4. The latter revealed that the discrepancy between nucleosome depletion in vivo and in vitro appears more pronounced also in MNase-seq data, so that we changed the respective paragraph accordingly (lines 320-332: “The discrepancy between nucleosome depletion in vivo versus in vitro was much more pronounced over promoter regions (Figure 2) than over genome-wide poly(dA) tracts (Figure 3) for both MNase-seq and ODM-seq measurements. This was not due to the 10-18% of promoters without poly(dA) tracts, where nucleosome depletion has to occur by a poly(dA)-independent mechanism in vivo, as this was still seen if we focused on  poly(dA) tracts and compared promoter versus non-promoter regions (Figure 4). As this was still seen also for MNase-seq, which even inflates the intrinsic depletion over poly(dA) tracts, this suggests that there are mechanisms in vivo especially at promoters that enhance nucleosome depletion over poly(dA) tracts relative to the intrinsic mechanism in vitro (see below). In addition, this may reflect the influence of the known poly(dA)-independent mechanisms, like factor binding competition [68], at promoters. Also in Figure 4, nucleosome depletion in vitro was in all cases again much less pronounced than in vivo if monitored by ODM-seq.”)
